# Green Biosynthesis of Silver Nanoparticles Using *Eriobotrya japonica* (Thunb.) Leaf Extract for Reductive Catalysis

**DOI:** 10.3390/ma12010189

**Published:** 2019-01-08

**Authors:** Chen Yu, Jingchun Tang, Xiaomei Liu, Xinwei Ren, Meinan Zhen, Lan Wang

**Affiliations:** 1College of Environmental Science and Engineering, Nankai University, 38 Tongyan Road, Jinnan District, Tianjin 300350, China; yacerola@163.com (C.Y.); 1120160147@mail.nankai.edu.cn (X.L.); renxinwei59@163.com (X.R.); zhenmeinan@163.com (M.Z.); envwangl@nankai.edu.cn (L.W.); 2Key Laboratory of Pollution Process and Environmental Criteria (Ministry of Education), College of Environmental Science and Engineering, Nankai University, Tianjin 300350, China; 3Tianjin Engineering Center of Environmental Diagnosis and Contamination Remediation, College of Environmental Science and Engineering, Nankai University, Tianjin 300350, China

**Keywords:** green biosynthesis, silver nanoparticles, *Eriobotrya japonica*, plant extract, catalyst

## Abstract

This article reports on silver nanoparticles (AgNPs) that were green-synthesized by using *Eriobotrya japonica* (Thunb.) leaf extract and their use for the catalytic degradation of reactive dyes. The properties of biogenic AgNPs were characterized using UV-vis absorption spectroscopy, field emission scanning electron microscope (FESEM), X-ray powder diffraction (XRD), transmission electron microscope (TEM), Fourier transforming infrared spectroscopy (FTIR), energy dispersive X-ray spectroscopy (EDS), and selected area electron diffraction (SAED) analysis. The UV-vis spectroscopy and X-ray analyses confirmed the formation of AgNPs and showed the strong absorbance around 467 nm with surface plasmon resonance (SPR). The mean diameter of biogenic AgNPs at room (20 °C), moderate (50 °C), and high temperatures (80 °C) were 9.26 ± 2.72, 13.09 ± 3.66, and 17.28 ± 5.78 nm, respectively. The reaction temperature had significant impacts on the sizes of synthesized AgNPs. The higher the synthesis temperature, the larger size and the lower catalysis activity for reductive decomposition of reactive dyes via NaBH_4_. The results supported a bio-green approach for developing AgNPs with a small size and stable degradation activity of reactive dyes over 92% in 30 min by using *Eriobotrya japonica* (Thunb.) leaf extract at pH 7, 20 °C, and 1:10 ratio of silver nitrate added to the leaf extract.

## 1. Introduction

Nanotechnology is an area of the most promising leading science in modern key technology development. Due to the combination of novel knowledge from chemistry, biology, materials science, and other related branches of science, the synthesis of nanoparticles has a wide and significant range of applications in catalysis, optics, biomedicine, and energy [1]. Nanoparticles can be synthesized using various traditional chemical and physical processes; however, they are characterized by low stability, the toxicity of chemicals, and the difficulty in controlling the growth of crystals and the aggregation [2,3]. Mirtaheri et al. (2017) synthesized mesoporous tungsten oxide using a template-assisted sol-gel method for the photocatalytic degradation of Rhodamine B [4]. Haghighatzadeh et al. (2017) synthesized mesoporous TiO_2_-SiO_2_ via an ultrasonic impregnation method and the anatase crystals synthesized under 800 °C with a higher photocatalytic efficiency for methylene blue degradation [5]. By comparison, green biosynthetic methods are much safer, greener, and eco-friendlier approaches [6]. The biogenic synthesis of noble metal nanoparticles as the burgeoning green nanotechnology has profound significance.

Nowadays, silver nanoparticles (AgNPs) have attracted much attention due to their fascinating properties, such as the good electrical and thermal conductivity, photo-electrochemical activity, chemical stability, and high catalytic and antimicrobial activities [7,8,9,10]. Moreover, variable sizes and shapes of AgNPs synthesized by biological agents, such as bacteria, algae, fungi, plants, and their enzymes and extracts, could have different mechanical, electrical, and structural properties [11,12,13,14]. Biosynthesized AgNPs (30–60 nm) using an isolated bacterium showed higher catalytic activity and stability towards the oxidation reaction of hydrazine [15]. Abdel-Raouf et al. (2018) synthesized AgNPs (39.41–77.71 nm) using the marine red alga *Laurencia catarinensis* using a rapid biogenic process (2 min–3 h) [16]. Arunachalam et al. (2012) used the leaf extract of *Coccina grandis* to successfully synthesize AgNPs with suitable activity for the degradation of Coomassie Brilliant Blue under UV light [17]. Extracts of plants and their seeds contain various natural antioxidants, generally, these biomolecules and metabolites can act as reducing and capping agents for biogenic synthesis of metal nanoparticles [18,19]. Plant extract-mediated synthesis of AgNPs are significant for its rapid rate of synthesis, simplicity, and eco-friendliness. *Erioboytrya japonica* (Thunb.), commonly know as loquat, is a multipurpose plant and its leaves have been used as herbal medicine to treat coughing due to the active chemical constituents such as flavonoids, triterpenoid acid, and sesquiterpene glycosides [20,21]. There is an urgent need to develop the environmental-friendly methods of green synthesis of AgNPs that do not use toxic chemicals. Therefore, the biomolecules and metabolites in *Erioboytrya japonica* (Thunb.) leaves are considered as suitable sources for the reducing and capping agents that could avoid agglomeration by stabilizing nanoparticles [22].

The present study investigated the properties of *Eriobotrya japonica* (Thunb.) leaf extract as the stabilizer and reducing agents for the synthesis of silver nanoparticles. The aim of this work was to employ a simple method for optimizing biosynthesis of small AgNPs and investigate their physicochemical properties. To the best of our knowledge, this is the first report for the biosynthesis of AgNPs using the *Eriobotrya japonica* (Thunb.) leaf extract and its application in the catalytic degradation of reactive dyes (e.g., Reactive Black 5 and Reactive Red 120) with NaBH_4_.

## 2. Materials and Methods

### 2.1. Preparation of Leaf Extract and Chemicals

*Eriobotyra japonica* (Thunb.) leaves were collected from Yangshan County, Guangdong Province, China (23°25′21.74″ N 113°03′24.25″ E). The leaves were washed several times with distilled water, shade dried for 3 days in the air, and then crushed into a good powder using the electronic blender. Thirty grams of the leaf powder was stirred with 500 mL of Milli-Q water and kept at 65 °C for 0.5 h. Then the extracts were filtered by using Whatman No. 1 filter paper after cooling to room temperature. The extract was stored at 4 °C for future use. Silver nitrate (AgNO_3_, 99.8%) and sodium borohydride (NaBH_4_, 98%) were procured from Aladdin Industrial Corporation (Shanghai, China). Reactive Red 120 and Reactive Black 5 were purchased from Sigma-Aldrich (Shanghai, China).

### 2.2. Green Biosynthesis of Ag Nanoparticles

The silver nanoparticles were prepared using 1mM silver nitrate solution, which was slowly mixed with the leaf extracts in a magnetic stirrer (Figure 1). The evidence of reduction was indicated by a rapid color change from light yellow to dark brown within 5 min [23]. The effect of different preparing conditions on the synthesis of AgNPs is indicated in Table 1 using an orthogonal experimental design: (1) temperature, at room temperature (G-L, 20 °C), moderate temperature (G-M, 50 °C), and high temperature (G-H, 80 °C); (2) pH (7.0, 7.5 and 8.0); and (3) different ratios of leaf extract and silver salt solution (1:1, 1:2, and 1:10, *v*/*v*). The silver nanoparticle solution was purified in Milli-Q water by thrice centrifugation at 10,000 rpm for 20 min.

### 2.3. Characterization Method and Instrument

The UV-visible spectra were recorded on a Presee TU-1950 spectrophotometer (Purkinje, Beijing, China) at a resolution of 1 nm to scan the samples in a wavelength range from 200 to 900 nm. A JSM-7800F field emission scanning electron microscope (FESEM) (JEOL, Tokyo, Japan) was used to image and study the size and morphology of AgNPs. The morphology, size, and electron diffraction pattern (SAED) of the silver nanoparticles were imaged using a JEM-2800 transmission electron microscope (TEM) (JEOL, Tokyo, Japan) with an accelerating voltage of 300 kV. X-ray diffraction pattern (XRD) images of dry nanoparticle powder was obtained using an Ulitama IV X-ray diffractometer (Rigaku, Tokyo, Japan) at the angle range of 2θ (10–80°) [24]. The Fourier transfer infrared (FTIR) spectra were obtained on a Bruker Tensor 37 (Bruker-AXS GmbH, Karlsruhe, Germany) FTIR instrument. The elemental compositions were detected using energy dispersive X-ray spectrometer (EDS) using x-act with INCA^®^ and Aztec^®^ EDS analysis software (Oxford Instruments, London, UK).

### 2.4. Catalytic Activity of AgNPs

To evaluate the catalytic activity of synthesized silver nanoparticles, the degradation of Reactive Red 120 and Reactive Black 5 was evaluated in aqueous solution as a model system in the presence of NaBH_4_ [25]. One milliliter of AgNPs (0.1 mM) solution and 0.5 mL NaBH_4_ (0.1 M) were added in a 5 mL reactive dye (one of either Red 120 or Black 5) solution (50 mg/L). Meanwhile, the catalytic performance of biogenic AgNPs was monitored by recording the UV-vis spectra in 30 min.

The reaction kinetics was evaluated by assuming the concentration of reactive dyes obeying the pseudo-first order reaction, where the integrated form is expressed as follows:$$\ln\frac{A_{t}}{A_{o}}=-kt$$
where *A*_0_ is the absorbance at zero time, *A*_t_ is the absorbance at *t* time and *k* is the rate constant.

## 3. Results and Discussion

### 3.1. Confirmation of AgNPs Formation using UV-Visible Spectroscopy Analysis

The biogenic nanoparticles were primarily characterized by the rapid color change of *Eriobotyra japonica* (Thunb.) leaf extract solution from light yellow to dark brown within 5 min after silver nitrate was added (Figure 1). The UV-vis spectra showed strong evidence of colloidal metal particle formation, and the productivity growth in the synthesis medium was indicated by the gradual increase in the absorbance values [26,27]. The silver nanoparticles have free electrons that stimulate the surface plasmon resonance absorption (SPR) band. Figure 2 shows the UV-vis spectra of AgNPs recorded under varying reaction conditions in the wavelength range from 200 to 900 nm and the maximum absorbance of silver nanoparticles occurred at 469 nm. The appearance of the peaks, assigned to surface plasmons, are well-documented for various metal nanoparticles with sizes ranging from 2 to 100 nm [17]. In general, AgNPs showed the absorption spectrum under SPR in the range of 450–490 nm. The sharp peaks indicate the same shape of nanoparticles, while the wide shapes mean that non-uniform particles shapes were created such as triangular and hexagonal shapes. The blue shift was detected in G-H2 due to the SPR oscillation of biogenic AgNPs [28].

### 3.2. TEM, FESEM, and SAED Analyses

Figure 3 shows the average particle sizes in different treatments based on the TEM results. By analyzing using SPSS 19.0 (IBM, New York, NY, USA), the relative magnitude of reaction conditions on synthesized effect was found (Table 2). The reaction temperature had a remarkable effect on the bio-synthesis of AgNPs compared with the ratio rate and pH. The relationship between particle size and temperature is shown in Figure 3a. The average diameter of synthesized AgNPs increased with the higher reaction temperature. The growth of bigger AgNPs was probably due to the faster nucleation of seed particles under higher temperature condition, subsequently. As shown in Figure 3b–d, the smaller mean diameter of biogenic AgNPs under different temperature conditions synthesized using G-L1 (20 °C), G-M1 (50 °C), and G-H2 (80 °C) were 9.26 ± 2.72, 13.09 ± 3.66, and 17.28 ± 5.78 nm, respectively.

Morphological analysis was conducted using FESEM and TEM as shown in Figure 3. FESEM analysis was carried out to understand the monodisperse spherical shape of the biogenic AgNPs (Figure 3e) with sizes ranging between 3–30 nm, which was also confirmed using TEM images (Figure 3f). The lattice spacing measured using HR-TEM (high resolution transmission electron microscopy) was 0.24 nm (Figure 3g). The obtained values of d-spacing between the lattice of Ag particles were in good agreement with Ag (111) and (220) as reported [29]. The SAED pattern of the Ag nanoparticles is shown in Figure 3h, where the ring-like diffraction pattern could be indexed on the basis of the faced-centered cubic (fcc) structure of silver. Four bright rings correlated to the (111), (200), (220), and (311) lattice planes of fcc silver, which confirmed that biosynthesized AgNPs had good crystallinity. 

### 3.3. XRD, FTIR, and EDS Analysis of Biogenic AgNPs

Analysis of silver nanoparticles using XRD pattern confirmed the purity and crystalline of the synthesized AgNPs (Figure 4a). In the XRD spectra, the strong diffraction peaks at 2*θ* = 38.03°, 46.18°, 63.43° and 77.18° correspond to the diffraction planes of (111), (200), (220) and (311), respectively. The sharp and narrow diffraction peaks in the XRD spectrum were in agreement with SAED analysis indicated that the synthesized AgNPs were pure and highly crystalline nature [30]. FTIR analysis indicated the possible functional groups responsible for reduction and stabilization of AgNPs synthesized by using *Eriobotyra japonica* (Thunb.) leaf extract (Figure 4b). FTIR spectrum of biogenic silver nanoparticles was in the range of 1064–3716 cm^−1^. The peaks appearing at 2921.49, 1621.99, and 1064.48 cm^−1^ in the FTIR spectrum of AgNPs were attributed to the O–H stretching, N–H stretching, and the stretching vibrations of the C=O functional groups of aldehydes, ketones, and carboxylic acids [31,32], respectively. Moreover, the FTIR spectrum of the leaf extract showed distinct peaks from 612 to 3313 cm^−1^. In the FTIR spectrum of the leaf extract, the peaks observed at 3313.49 and 1376.69 cm^−1^ were due to the C–H stretching of alkanes. The band at 1736 cm^−1^ is attributed to be the C=O stretching [33] whereas a peak at 780.32 cm^−1^ observed in the FTIR spectrum was attributed to N–H stretching. The strong peaks at 1317.34 and 612.38 cm^−1^ indicated the presence of N–O symmetric stretching and alkyl halides [34,35], respectively. In the biogenic synthesis of AgNPs, N–H and carbonyl functional groups containing amides were involved in the reduction of silver ions to silver nanoparticles with the addition of *Eriobotyra japonica* (Thunb.) leaf extract to silver nitrate solution. Additionally, EDS analyses confirmed the presence of silver nanoparticles via further analysis.

Figure 4c depicts the identification of AgNPs synthesized at 20, 50, and 80 °C using *Eriobotyra japonica* (Thunb.) leaf extraction without any impurity peaks. Silver nanocrystallites displayed an optical absorption band peak at approximately 3 keV, which is typical of the absorption of metallic silver nanocrystals due to surface plasmon resonance [36]. Therefore, the biogenic silver nanoparticles were successfully fabricated with less impurity. Elements C, O, and N were also found in the EDS spectrogram, indicating the presence of organic components in the biogenic nano-materials at the same time. The EDS spectra clearly showed that with a decrease in the temperature, the percentage of AgNPs increased from 64.7% to 84.9% in the nanomaterials, while the percentage of elements of C, O, and N decreased with the decrease in temperature. This result indicated that the biogenic AgNPs with high purity could be synthesized at room temperature. The leaf extract has a variety of metabolites such as organic acids, terpenoids, flavonoids, etc. These phytocompounds are responsible for the reducing property that acids have in the immediate reduction of silver ions into silver nanoparticles and stabilizing AgNPs to prevent agglomeration as a capping agent. In this study, we successfully synthesized Ag nanoparticles with a small size at pH 7 and 20 °C, and the proportion ratio of leaf extract to silver nitrate was 1:10 (*v*:*v*).

### 3.4. Reductive Degradation of Reactive Dyes by Biogenic AgNPs

#### 3.4.1. Compare of Degradation Ability of Different Catalysts

Reactive Red 120 and Reactive Black 5 are azo dyes and are widely used for the dyeing of cellulosic yarns and fabrics, but it is non-biodegraded by conventional activated sludge processes. Reactive Red 120 shows strong absorbance peaks at 536nm, as shown in Figure 5a–d, and the strong absorbance peaks of Reactive Black 5 are shown in Figure 5e–h at 594 nm. The two reactive dyes were minimally degraded by NaBH_4_ when AgNPs were not added (Figure 5a,e). The absorbance of the reactive dyes at strong absorbance peaks remained unchanged over a period of 30 min, which indicated that Reactive Red 120 and Reactive Black 5 could not be degraded individually by NaBH_4_. With the addition of AgNPs to the mixture containing reactive dyes and NaBH_4_, an immediate decrease occurred and then continued for the reaction time of 2 min. The synthesized size of silver nanoparticles had a great impact on the catalytic activity for reductive decomposition of reactive dyes. It was observed that the higher the synthesis temperature, the larger the size of AgNPs; however, the reaction rate of dye degradation was slower. The imparity in degradation activity should be associated with the increase size of biogenic nanoparticles and less surface areas when AgNPs are synthesized under higher temperature. Due to the finer AgNPs having greater surface areas and faster electron transport, they could rapidly break the two azo bonds of reactive dyes when forming an intermediate such that the amino groups could be further decomposed. The results supported a bio-green approach for developing silver nanoparticles with small sizes and stable degradation activity of reactive dyes by using *Eriobotrya japonica* (Thunb.) leaf extract at room temperature (20 °C).

#### 3.4.2. Evaluation of Reductive Degradation Activity of Reactive Dyes

As shown in Figure 6, the dye degradation process is assumed to follow a pseudo-first order law: $\ln\frac{A_{t}}{A_{0}}=-kt$, where *A*_0_ and *A*_t_ are the absorbance at zero time and time *t*, respectively, and *k* is the rate constant. The dye degradation of Reactive Red 120 using synthesized AgNPs by G-L1, G-M1, and G-H2 were calculated to be 92.01%, 83.54%, and 81.27%, respectively. While the biogenic AgNPs had better degradation capacities toward Reactive Black 5 of 93.98%, 94.76%, and 78.01% in 30 min catalyzed by G-L1, G-M1, and G-H2, respectively. It was observed that degradation rates of the reactive dyes decreased with increasing processing temperature. The degradation rate of Reactive Red 120 constantly decreased from 0.0802 min^−1^ (for G-L1) to 0.032 min^−1^ (for G-H2) when AgNPs were synthesized under high-temperature conditions. Meanwhile, the constant decreased from 0.1259 min^−1^ to 0.0477 min^−1^ in the degradation process of Reactive Black 5. The synthesized size of nanoparticles had a significant impact on the catalytic activity of biogenic silver nanoparticles for their reductive decomposition of reactive dyes using NaBH_4_. Furthermore, it is certain that the biosynthesis process of silver nanoparticles under high-temperature conditions have impaired impacts on catalytic properties in comparison with the AgNPs synthesized under mild temperature conditions.

## 4. Conclusions

The study demonstrated that AgNPs can be green-synthesized by using *Eriobotyra japonica* (Thunb.) leaf extract. The leaf extract could act as reducing and capping agents in the eco-friendly and green synthesis process. The UV-vis spectroscopy and X-ray analyses confirmed the formation of biogenic AgNPs, which showed the strong absorbance at 469 nm with surface plasmon resonance. The synthesized size of silver nanoparticles had great impacts on the catalytic activity for the reductive decomposition of reactive dyes. It was found that the higher the synthesis temperature, the larger the size and the slower the reaction rate. The biogenic AgNPs with a small mean diameter about 9.26 ± 2.72 nm could be synthesized at pH 7 and 20 °C, and the proportion ratio of *Eriobotyra japonica* (Thunb.) leaf extract and silver nitrate was 1:10 (*v*:*v*). The results supported a bio-green approach for developing silver nanoparticles with small sizes and the stable degradation activity of reactive dyes by using *Eriobotyra japonica* (Thunb.) extract at room temperature.

## Figures and Tables

**Figure 1 materials-12-00189-f001:**
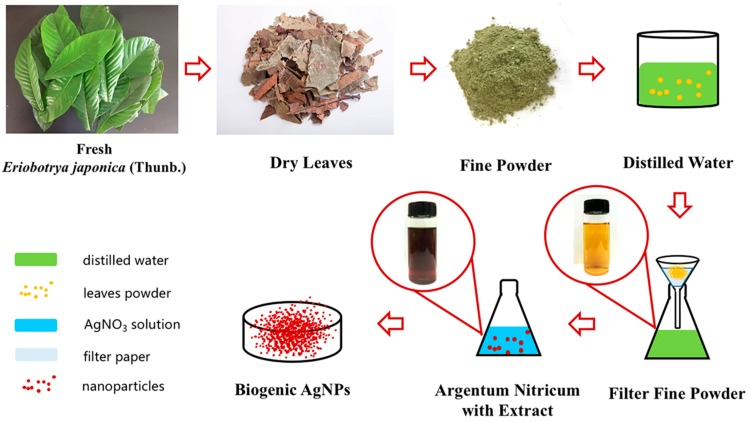
Schematic depiction of the preparation of AgNPs using *Eriobotrya japonica* (Thunb.).

**Figure 2 materials-12-00189-f002:**
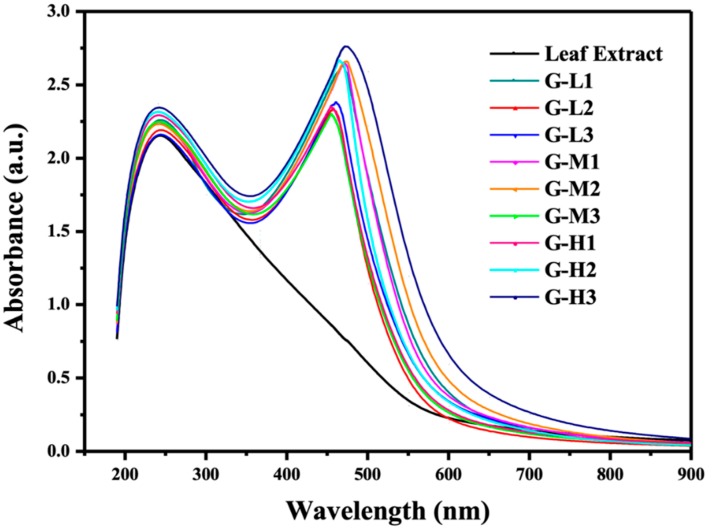
UV–vis spectra of *Eriobotrya japonica* (Thunb.) leaf extract and AgNPs solutions prepared at varying reaction conditions.

**Figure 3 materials-12-00189-f003:**
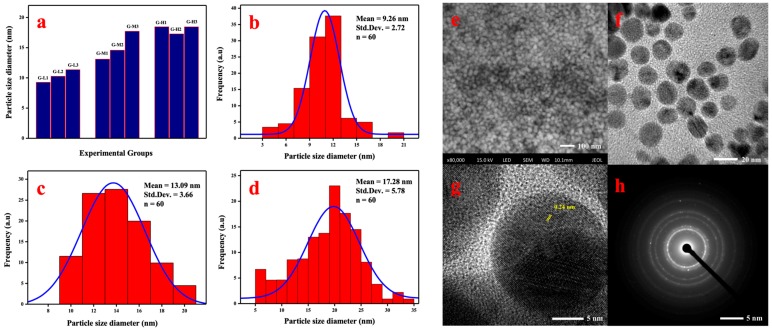
Particle sizes of synthesized AgNPs on different treatment conditions (**a**), and frequency distribution histograms of AgNPs synthesized by G-L1 (**b**), G-M1 (**c**), and G-H2 (**d**). SEM images (**e**), TEM images (**f**), the electron diffraction pattern of AgNPs using the HR-TEM image (**g**), and SAED pattern (**h**) of AgNPs synthesized using G-L1.

**Figure 4 materials-12-00189-f004:**
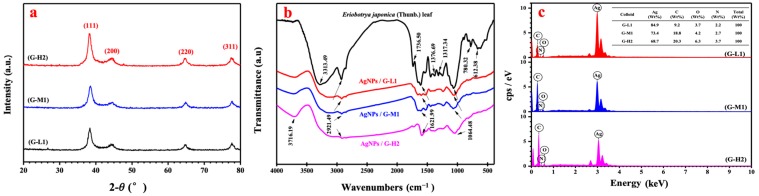
X-ray diffraction pattern (**a**), the FTIR spectra (**b**) of dried leaf powder and biogenic AgNPs and EDS analysis (**c**) of synthesized silver nanoparticles by G-L1, G-M1 and G-H2.

**Figure 5 materials-12-00189-f005:**
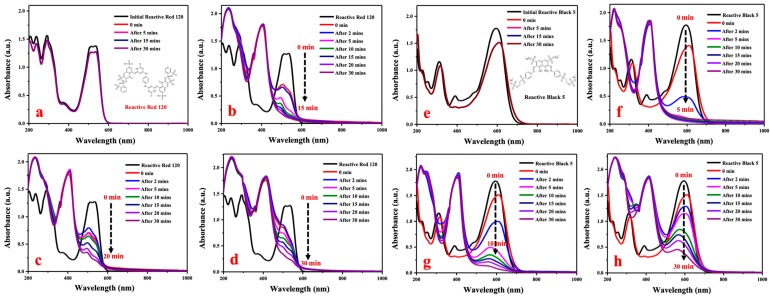
Reductive degradation of Reactive Red 120 using different catalysts: (**a**) No AgNPs, (**b**) G-L1, (**c**) G-M1, and (**d**) G-H2, and reductive degradation of Reactive Black 5 using (**e**) No AgNPs, (**f**) G-L1, (**g**) G-M1, and (**h**) G-H2.

**Figure 6 materials-12-00189-f006:**
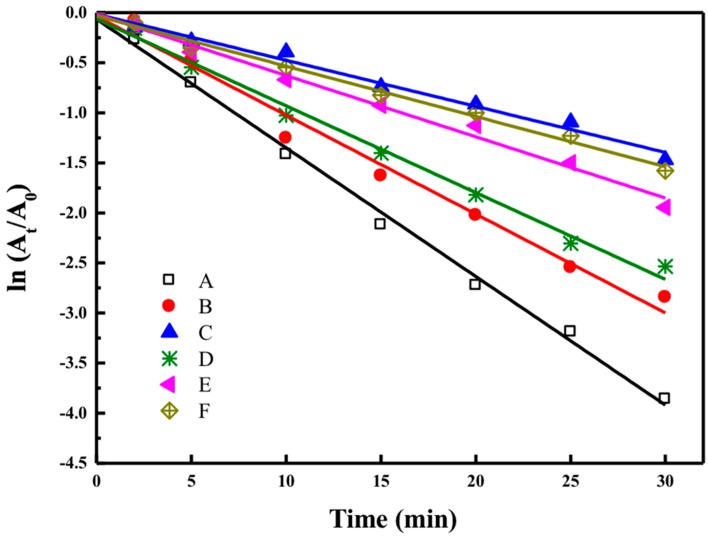
The ln(A_t_/A_0_) versus time linear fits during the degradation of Reactive Black 5: (**A**) G-L1, (**B**) G-M1, and (**C**) G-H2, and Reactive Red 120: (**D**) G-L1, (**E**) G-M1, and (**F**) G-H2.

**Table 1 materials-12-00189-t001:** AgNPs biosynthesis parameters.

Colloid	Temperature	Proportion Ratio of Silver Nitrate and Leaf Extract (*v*:*v*)	pH
	(°C)	(mL)	
G-L1	20	10:1	7.0
G-L2	20	2:1	7.5
G-L3	20	1:1	8.0
G-M1	50	10:1	7.5
G-M2	50	2:1	8.0
G-M3	50	1:1	7.0
G-H1	80	10:1	8.0
G-H2	80	2:1	7.0
G-H3	80	1:1	7.5

G: Grouping experiments.

**Table 2 materials-12-00189-t002:** The statistical analysis (*p* < 0.05).

Influencing Factor	DEVSQ	DOF	*F Value*	*F* Critical-Value	Significance
Temperature (°C)	91.436	2	63.016	5.140	*
Proportion ratio of silver nitrate and leaf extract (*v*:*v*)	1.622	2	1.000	5.140	
pH	0.228	2	1.118	5.140	

*: significant effect.

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
