# Peer review of "Green Biosynthesis of Silver Nanoparticles Using *Eriobotrya japonica* (Thunb.) Leaf Extract for Reductive Catalysis"

_materials, 2019, doi:10.3390/ma12010189_

Round 1

Reviewer 1 Report

This paper describes a green synthesis of Ag nanoparticles with the use of a plant leaf extract, testing different operative temperatures and pH values. The behaviour in the degradation of Reactive Red 120 and Reactive Black 5 dyes has been evaluated.

To be considered for publication, the paper needs to be improved. The english level is quite low, and many parts of the papers are not clear and not well described. The introduction section needs to be enhanced in terms of background and references.

1) There are many grammar mistakes and the whole paper needs a deep check and improvement of the english level.

2) Line 128: SPR means “Surface Plasmon Resonance” not Pasmon.

3)Regarding the UV vis spectra of samples, the authors should better explain and justify the shift of the band in the 450-490nm range. Do the dimensions of the nanoparticles are involved?

4)Concerning EDS characterization (Fig. 4c), why do the author refer to the peak of Ag nanocristals as an “optical absorption band?” (line 190)?

Author Response

Response to Reviewer 1 Comments

ManuscriptNo.:  materials-408025

Title:Green biosynthesis of silver nanoparticles using Eriobotrya japonica(Thunb.) leaf extract for reductive catalysis

Corresponding author:Dr. Jingchun Tang

Note: Pageand line numbers in the response refer to those in the revisedversion of the manuscript unless indicated otherwise.

Reviewer 1:This paper describes a green synthesis of Ag nanoparticles with the use of a plant leaf extract, testing different operative temperatures and pH values. The behaviour in the degradation of Reactive Red 120 and Reactive Black 5 dyes has been evaluated. To be considered for publication, the paper needs to be improved. The english level is quite low, and many parts of the papers are not clear and not well described. The introduction section needs to be enhanced in terms of background and references. 

Authors: Thank you for the thorough review and helpful comments. Please refer to the following point-by-point responses to the specific comments.

Specific comments:

Point 1:There are many grammar mistakes and the whole paper needs a deep check and improvement of the english level.

Response 1:Thank you for your suggestion. Recheck and corrected the definite and indefinite articles on the right positions.

(Line 39): “...such as the good electrical and thermal conductivity,”

(Line 81): “... sliver nanoparticles were prepared using 1mM sliver nitrate solution...”

(Line 135): “...the absorption spectrum under SPR in the range of 450-490 nm.”

The wrong spelling has been corrected. 

(Line 129-130): “The silver nanoparticles have free electrons that stimulate the surface plasmon resonance absorption (SPR) band.”

(Line 146):“...temperature had remarkable effect on bio-synthesis of AgNPs...”

The completeterms of the missing supplement before the first use for abstract and main manuscript.

(Line 10):“...467 nm with surface plasmon resonance (SPR).” 

(Line 103-104): “X-ray diffraction pattern (XRD) of dry nanoparticle powder was obtained using Ulitama IV X-ray diffractometer (Rigaku, Tokyo, Japan) at the angle range of 2θ (10-80°).”

(Line 160-161): “The lattic spacing measured by HR-TEM (high resolution transmission electron microscopy) was 0.24 nm (Fig. 3g).”

Point 2:Line 128: SPR means “Surface Plasmon Resonance” not Pasmon.

Response 2:Thank youfor your suggestion. The wrong spelling has been corrected.

(Line 129-130): “The silver nanoparticles have free electrons that stimulate the surface plasmon resonance absorption (SPR) band.”

Point 3:Regarding the UV vis spectra of samples, the authors should better explain and justify the shift of the band in the 450-490 nm range. Do the dimensions of the nanoparticles are involved?

Response 3: Thank you very much for your valuable suggestions. UV-vis spectroscopy reveals the SPR of the Ag electrons and offers information regarding the size and shape of the nanoparticles by an explanation about blue shift (shift to a low wave-length which means the size of particle decrease) and red shift (shift to a high wavelength which mean the size of particle increase). The wide peak shows that there is a different shape of particles while the sharp peak displays the same shape of nanoparticles in the range of 450-490 nm.

Point 4:Concerning EDS characterization (Fig. 4c), why do the author refer to the peak of Ag nanocrystals as an “optical absorption band?” (line 190)?

Response 4: Thank you very much for your valuable suggestions. Due to the surface plasmon resonance (SPR), the biogenic silver nanoparticles display an optical absorption band peaked at about 3 KeV (410 nm), which is the typical of absorption at metallic Ag nanoclusters. So the absorption band is consistent with the value for Ag nanoclusters. The related reference is listed in Line 199 and Line 375-376.

Reviewer 2 Report

This work presents Green biosynthesis of silver nanoparticles using Eriobotrya japonica leaf extract: Characterization and application for reductive catalysis. The nanoparticles synthesis and applications achieved in this research are important and interesting. It is recommended to be published after including and addressing the below listed comments with major corrections.

- The authors should eliminate the current grammatical and punctuation mark errors and also confirm the correct scientific English. Make sure to avoid or insert commas on the right positions. Make sure to use the definite and indefinite articles on the right positions.

- The authors should write the complete terms of all abbreviations before the first use for both abstract and main manuscript.

The authors are suggested to revised the title of the manuscript, make it shorter and more meaningful.

- The introduction part of the manuscript needs more development to explain the novelty and importance of the reported work and including the supported catalysts while citing previously published articles.

- Enlarge the scale bars of the Figs. 

- Draw a reaction scheme or chemical structure of the chemicals used in the reactions if possible for the catalytic Figs.

- The authors should add the important review and experimental published papers for Pd nanocatalysts, supported catalysts and hydrogenation processes including the below mentioned references on their revised manuscript:

Desalination and Water Treatment 92, 145-151 (2017).

J. Sol. Gel Sci. Tech. 82, 148-156 (2017).

- Mention the reaction condition for all reaction plots and also recycling graph on the revised manuscript.
- The authors can provide more scientific catalytic studies such as reaction optimization, and elaborate their research.

Author Response

Response to Reviewer 2 Comments

Manuscript No.:  materials-408025

Title:Green biosynthesis of silver nanoparticles using Eriobotrya japonica(Thunb.) leaf extract for reductive catalysis

Corresponding author: Dr. Jingchun Tang

Note: Page and line numbers in the response refer to those in the revisedversion of the manuscript unless indicated otherwise.

Reviewer 2: This work presents Green biosynthesis of silver nanoparticles using Eriobotrya japonicaleaf extract: Characterization and application for reductive catalysis. The nanoparticles synthesis and applications achieved in this research are important and interesting. It is recommended to be published after including and addressing the below listed comments with major corrections.

Authors: Thank you for the thorough review and helpful comments. Please refer to the following point-by-point responses to the specific comments.

Specific comments:

Point 1:The authors should eliminate the current grammatical and punctuation mark errors and also confirm the correct scientific English. Make sure to avoid or insert commas on the right positions. Make sure to use the definite and indefinite articles on the right positions.

Response 1:Thank you for your suggestion. 

The wrong spelling has been corrected. 

(Line 129-130): “The silver nanoparticles have free electrons that stimulate the surface plasmon resonance absorption (SPR) band.”

(Line 146): “...temperature had remarkable effect on bio-synthesis of AgNPs...”

Rechecked and corrected the definite and indefinite articles on the right positions.

(Line 39): “...such as the good electrical and thermal conductivity,”

(Line 81): “... sliver nanoparticles were prepared using 1mM sliver nitrate solution...”

(Line 135): “...the absorption spectrum under SPR in the range of 450-490 nm.”

Point 2:The authors should write the complete terms of all abbreviations before the first use for both abstract and main manuscript

Response 2:Thank you for your valuable suggestion. The complete terms of the missing supplement before the first use for abstract and main manuscript.

(Line 10): “...467 nm withsurface plasmon resonance (SPR).” 

(Line 103-104):X-ray diffraction pattern (XRD) of dry nanoparticle powder was obtained using Ulitama IV X-ray diffractometer (Rigaku, Tokyo, Japan) at the angle range of 2θ (10-80°).”

(Line 160-161):The lattic spacing measured by HR-TEM (high resolution transmission electron microscopy) was 0.24 nm (Fig. 3g).”

Point 3:The authors are suggested to revised the title of the manuscript, make it shorter and more meaningful.

Response 3:Thank you for your valuable suggestion.The title was changed as:

Green biosynthesis of silver nanoparticles using Eriobotrya japonica(Thunb.) leaf extract for reductive catalysis”.

Point 4:The introduction part of the manuscript needs more development to explain the novelty and importance of the reported work and including the supported catalysts while citing previously published articles

Response 4:Thank you very much for your valuable suggestions. For better emphasizedthesignificance of biosynthesis of AgNPs by use of Eriobotrya japonica (Thunb.) leaf extract. Here are the changes

(Line 56-60):“There is an urgent need to develop the environmental-friendly methods of green synthesis of AgNPs that do not use of toxic chemicals. Therefore, the biomolecules and metabolites in Erioboytrya japonica(Thunb.) leaves are considered as the suitable sources for the reducing and capping agents that could avoid agglomeration by stabilizing nanoparticles.”

References with catalysts are cited:

(Line 45-49):Abdel-Raouf et al. (2018) synthesized AgNPs (39.41-77.71 nm) using the marine red alga Laurencia catarinensis with rapid biogenic process (2 min-3 h). Arunachalam et al. (2012) used the leaf extract of Coccina grandisto successfully synthesize AgNPs with suitable activity for the degradation of Coomassie Brilliant Blue under UV light.”

Point 5:Enlarge the scale bars of the Figs.

Response 5:Thank you very much for your valuable suggestions. To make the Figs more coordinate and beautiful, the scale bars of TEM images have been modified.

Point 6:Draw a reaction scheme or chemical structure of the chemicals used in the reactions if possible for the catalytic Figs.

Response 6:Thank you very much for your valuable suggestions. The chemical structure of Reactive Red 120 and Reactive Black 5 have been added to the figures.

Point 7:The authors should add the important review and experimental published papers for Pd nanocatalysts, supported catalysts and hydrogenation processes including the below mentioned references on their revised manuscript:

Desalination and Water Treatment 92, 145-151 (2017).

J. Sol. Gel Sci. Tech. 82, 148-156 (2017).

Response 7:Thank you very much for your valuable suggestions. The mentioned references have been added to the revised manuscript.

(Line 30-31): “Mirtaheri et al. (2017) synthesized mesoporous tungsten oxide by templated assisted sol-gel method for the photocatalytic degradation of Rhodamine B.”

(Line 32-34): “Haghighatzadeh et al. (2017) synthesized mesoporous TiO2-SiO2via ultrasonic impregnation method and the anatase crystals synthesized under 800℃ with higher photocatalytic efficiency for methylene blue degradation”

Point 8:Mention the reaction condition for all reaction plots and also recycling graph on the revised manuscript.

Response 8:Thank you very much for your valuable suggestions. The reaction conditions for all reaction plots have been listed in Table 1. (Line 97-99)

The relevant text in the manuscriptwas re-explained to read:

(Line 84-88) : “The effect of different preparing conditions on the synthesis of AgNPs was indicated in Table 1 by orthogonality experimental design: (1) temperature, at room temperature (G-L, 20℃), moderate temperature (G-M, 50℃) and high temperature (G-H, 80℃), (2) pH (7.0, 7.5 and 8.0) and (3) different ratios of leaf extract and silver salt solution (1:1, 1:2 and 1:10, v/v).”

Point 9:The authors can provide more scientific catalytic studies such as reaction optimization and elaborate their research.

Response 9:Thank you very much for your suggestions. In order to study the effect of different preparing conditions on the synthesis of nanoparticles, we designed nine experimental groups and know the reaction temperature had remarkable effect on bio-synthesis of silver nanoparticles compared with ratio rate and pH. In addition, the discussion also referred to the reports of related literatures.

Reviewer 3 Report

The current manuscript reports on the green synthesis of silver nanoparticles (Ag NPs) using Eriobotrya japonica leaf (Thunb.) extract as reducing and stabilizing agent. The authors claimed that synthesis temperature has high impact on Ag particle sizes that lead to different catalytic activity. The experimental data for the effect of temperature on particle sizes are not clear, the authors must convince the readership by giving better microscopy and spectroscopy evidences for change in particle sizes with temperature. This paper could be published in the Materials journal after major revisions and taking the following comments into account.

1.     The given explanation for the observed blue shift for surface plasmon peak is unclear. It is well known that blue shift reflects reduction in particle size. The authors should correlate the effect of synthesis temperature and particle size using surface plasmon peak position. Otherwise include low magnification TEM images of NPs synthesized at different temperatures corresponding to histograms in Figure 3.

2.     Provide the crystal sizes of Ag NPs synthesized at different temperatures based on XRD patterns.

3.     The given statistical error for particle sizes should be based on different batches but not from the same batch as mentioned figure 3. There is not much difference in particle size between 20 and 50 deg C synthesis temperature if errors are considered.

4.     Provide the reasons for relative increase in Ag to C ratio at low synthesis temperature (Figure 4c)

5.     Please comment on synthesis yield of Ag NPs at different temperatures.

6.     What are the effects of NaBH4 on catalytic degradation of dyes? Did the authors try degradation of dyes over Ag NPs without NaBH4?

7.     There is no point in mentioning 0, 5, 15 mins in legend in figure 5 if you are not including the corresponding spectra. Please revise this figure. It is difficult to correlate the mentioned time with the curve. For instance, figure 5f, 5 min pointing to blue curve (legend says that is 2 mins)

8.     Please mention the surface areas of the samples if you think surface area has a huge impact on the activity.

9.     Why do the authors measure higher difference for black 5 over red 120, between small and large size particles (figure 6)?

10.  Did the authors check the catalytic stability of these NPs?

11.  The discussion about the characterization techniques is repeated in abstract and in end part of Introduction (Line. 67 to 70)

Author Response

esponse to Reviewer 3 Comments

Manuscript No.:  materials-408025

Title:Green biosynthesis of silver nanoparticles using Eriobotrya japonica(Thunb.) leaf extract for reductive catalysis

Corresponding author: Dr. Jingchun Tang

Note: Page and line numbers in the response refer to those in the revisedversion of the manuscript unless indicated otherwise.

Reviewer 3: The current manuscript reports on the green synthesis of silver nanoparticles (Ag NPs) using Eriobotrya japonicaleaf (Thunb.) extract as reducing and stabilizing agent. The authors claimed that synthesis temperature has high impact on Ag particle sizes that lead to different catalytic activity. The experimental data for the effect of temperature on particle sizes are not clear, the authors must convince the readership by giving better microscopy and spectroscopy evidences for change in particle sizes with temperature. This paper could be published in the Materials journal after major revisions and taking the following comments into account.

Authors: Thank you for the thorough review and helpful comments. Please refer to the following point-by-point responses to the specific comments.

Specific comments:

Point 1: The given explanation for the observed blue shift for surface plasmon peak is unclear. It is well known that blue shift reflects reduction in particle size. The authors should correlate the effect of synthesis temperature and particle size using surface plasmon peak position. Otherwise include low magnification TEM images of NPs synthesized at different temperatures corresponding to histograms in Figure 3.

Response 1:Thank you very much for your valuable suggestions. UV-vis spectroscopy reveals the SPR of the Ag electrons and offers information regarding the size and shape of the nanoparticles by an explanation about blue shift (shift to a low wave-length which means the size of particle decrease) and red shift (shift to a high wavelength which mean the size of particle increase). The wide peak shows that there is a different shape of particles while the sharp peak displays the same shape of biogenic silver nanoparticles. The blue shift of G-H2 indicated the smaller biogenic AgNPs were synthesized than G-H1 and G-H3 at high temperature. The particle sizes of synthesized AgNPs on different treatment conditions (Fig.3a) 

Point 2:Provide the crystal sizes of Ag NPs synthesized at different temperatures based on XRD patterns.

Response 2:Thank you very much for your valuable suggestions. The relevant text in the manuscriptwas re-explained to read:The mean diameter of biogenic AgNPs at room (20℃), moderate (50℃) and high temperatures (80℃) were 9.26 ± 2.72, 13.09 ± 3.66 and 17.28 ± 5.78 nm, respectively, based on XRD patterns.

Point 3:The given statistical error for particle sizes should be based on different batches but not from the same batch as mentioned figure 3. There is not much difference in particle size between 20 and 50 deg C synthesis temperature if errors are considered.

Response 3:Thank you very much for your valuable suggestions. Firstly, as the experimental design, the average particle sizes in different nine treatments were analyzed based on the TEM results. While, the biogenic AgNPs synthesized by G-L1, G-M1 and G-H2 had smaller particle size, for better research the particle sizes were analyzed in three parallel results as shown in Fig.3b-d.

For milder reaction conditions and lower energy consumption, the synthesized temperature at 20℃ is greener and more significant than 50℃.

Point 4:Provide the reasons for relative increase in Ag to C ratio at low synthesis temperature (Figure 4c).

Response 4:Thank you very much for your valuable suggestions. The reasons for relative increase in Ag to C ratio at low synthesis temperature. The EDS result is most likely as a consequence of the surface desorption of bio-organic compounds present in nanoparticles. Thus, plant leaf extract-stabilized Ag nanoparticles are expected to be made up of molecules responsible for the reduction of metal ion and stabilizing particles in the solution at lower temperature. The high ratio of Ag and C may indicate the phytocompound is responsible for the reduction of Ag+to Ag0could be a thermally low stable compound that gets extracted in water.

Point 5:Please comment on synthesis yield of Ag NPs at different temperatures.

Response 5:Thank you very much for your valuable suggestions. It was observed during the experiments all the nine experimental groups could rapidly synthesized silver nanoparticles within 5 min. The evidence of reduction was indicated by color change from light yellow to dark brown rapidly within 5 min.

Point 6:What are the effects of NaBH4on catalytic degradation of dyes? Did the authors try degradation of dyes over Ag NPs without NaBH4

Response 6:Thank you very much for your valuable suggestions. Reactive Red 120 and Reactive Black 5 are azo dyes and used for the dyeing of cellulosic yarns and fabrics, but it is non-biodegradable by means of conventional activated sludge processes. During the catalytic degradation, NaBH4is the reducing agent so that the cleavages of two azo bonds can first occur while forming an intermediate. Then, the amino groups in the intermediate can be further decomposed. So finer AgNPs with greater surface areas may enhance the rate of dye degradation. 

However, the relatively small dye degradation over AgNPs without NaBH4. In order to be clearer, the reactive dyes were not degraded significantly by the colorimetric test.

Point 7:There is no point in mentioning 0, 5, 15 mins in legend in figure 5 if you are not including the corresponding spectra. Please revise this figure. It is difficult to correlate the mentioned time with the curve. For instance, figure 5f, 5 min pointing to blue curve (legend says that is 2 mins).

Response 7: Thank you very much for your suggestions. However, due to the reductive degradation of two reactive dyes was minimal by alone when no AgNPs were added. So, there are overlap between the corresponding spectra of 0, 5 and 15 mins (as shown in Fig.5a and 5e). These results indicated that Reactive Red 120 and Reactive Black 5 should not be degraded solely by NaBH4

The arrows pointing to the curves in Figure 5 have been corrected based on your valuable suggestion.

Point 8:Please mention the surface areas of the samples if you think surface area has a huge impact on the activity.

Response 8:Thank you very much for your valuable suggestions. This is probably because the small size of silver nanoparticles has the bigger specific surface area which is more conducive to catalytic degradation of reactive dyes. 

Point 9:Why do the authors measure higher difference for black 5 over red 120, between small and large size particles (figure 6)?

Response 9:Thank you very much for your valuable suggestions. The reason why the higher difference for Reactive Black 5 over Reactive Red 120 might be the different molecular weight and symmetry of two reactive dyes. Reactive Black 5 has smaller molecular weight and better symmetrical, so it’s more easily and quickly degraded than Reactive Red 120.

Point 10:Did the authors check the catalytic stability of these NPs?

Response 10:Thank you very much for your valuable suggestions. The biogenic silver nanoparticles have been stored to check the catalytic stability in one month for further study.

Point 11:The discussion about the characterization techniques is repeated in abstract and in end part of Introduction (Line. 67 to 70).

Response 11:Thank you very much for your valuable suggestions. The repeated part of characterization techniques in the end part if Introduction has been deleted.

Round 2

Reviewer 1 Report

After the modifications dome ti the maniscript, I think that the paper is now suitable For publication

Reviewer 2 Report

The authors have solved the shortcoming of the first submitted manuscript. They have also included the referee's suggestions and comments on the revised manuscript. It is suggested the manuscript will be published after approve of the editor.

Reviewer 3 Report

The authors have addressed all my concerns and the revised version can be accepted in Materials.